# A Fast Retrieval of Cloud Parameters Using a Triplet of Wavelengths of Oxygen Dimer Band around 477 nm

**Haklim Choi** [1,*], **Xiong Liu** [2], **Gonzalo Gonzalez Abad** [2], **Jongjin Seo** [3], **Kwang-Mog Lee** [1] **and Jhoon Kim** [4]

[1] Department of Astronomy and Atmospheric Sciences, Kyungpook National University, Daegu 41566, Korea; kmlee@knu.ac.kr

[2] Harvard-Smithsonian Center for Astrophysics, 60 Garden Street, Cambridge, MA 02138, USA; xliu@cfa.harvard.edu (X.L.); ggonzalezabad@cfa.harvard.edu (G.G.A.)

[3] Department of Astronomy and Atmospheric Sciences, University of Wisconsin, Madison, WI 53706, USA; jseo47@wisc.edu

[4] Department of Atmospheric Sciences, Yonsei University, Seoul 03722, Korea; jkim2@yonsei.ac.kr

* Correspondence: haklim84@knu.ac.kr; Tel.: +82-53-950-6360

**Abstract:** Clouds act as a major reflector that changes the amount of sunlight reflected to space. Change in radiance intensity due to the presence of clouds interrupts the retrieval of trace gas or aerosol properties from satellite data. In this paper, we developed a fast and robust algorithm, named the fast cloud retrieval algorithm, using a triplet of wavelengths (469, 477, and 485 nm) of the $O_2$–$O_2$ absorption band around 477 nm (CLDTO4) to derive the cloud information such as cloud top pressure (CTP) and cloud fraction (CF) for the Geostationary Environment Monitoring Spectrometer (GEMS). The novel algorithm is based on the fact that the difference in the optical path through which light passes with regard to the altitude of clouds causes a change in radiance due to the absorption of $O_2$–$O_2$ at the three selected wavelengths. To reduce the time required for algorithm calculations, the look-up table (LUT) method was applied. The LUT was pre-constructed for various conditions of geometry using Vectorized Linearized Discrete Ordinate Radiative Transfer (VLIDORT) to consider the polarization of the scattered light. The GEMS was launched in February 2020, but the observed data of GEMS have not yet been widely released. To evaluate the performance of the algorithm, the retrieved CTP and CF using observational data from the Global Ozone Monitoring Experiment-2 (GOME-2), which cover the spectral range of GEMS, were compared with the results of the Fast Retrieval Scheme for Clouds from the Oxygen A band (FRESCO) algorithm, which is based on the $O_2$ A-band. There was good agreement between the results, despite small discrepancies for low clouds.

**Keywords:** fast retrieval; cloud top pressure; cloud fraction; oxygen dimer; GEMS; FRESCO; VLIDORT



## 1. Introduction

Clouds play an important role as a reflector that causes change in the amount of reflected sunlight in the ultraviolet–visible (UV–Vis) region. Moreover, clouds significantly attenuate the polarization of the atmosphere [1–3]. Thus, clouds affect the accuracy of the retrieval of trace gases and aerosols from satellites. In particular, considering the spatial resolution of satellites, only 5–15% of the observed pixels correspond to clear sky conditions [4].

Consequently, it is essential to provide cloud information for the pixels where cloud exists. For example, the amount of ozone under cloud is corrected using the climatic value since the cloud serves as a shield preventing sunlight from penetrating the atmosphere under the cloud [5,6]. Therefore, it is necessary to obtain accurate cloud information to calculate the amount of ozone precisely.

In the past, the methods for observing the composition of trace gases in the atmosphere were developed based on the instruments onboard the low Earth orbit (LEO) satellites

such as the Ozone Monitoring Instrument (OMI [7]), Global Ozone Monitoring Experiment (GOME [8]), GOME-2 [9,10], and Scanning Imaging Absorption Spectrometer for Atmospheric Chartography (SCIAMACHY [11]). These measure the solar radiance that is reflected by the Earth at the top of the atmosphere (TOA) to monitor the atmospheric composition of interest. In recent years, many organizations have contributed to the development of geostationary Earth orbit (GEO) satellites such as Tropospheric Emissions: Monitoring of Pollution (TEMPO) [12], Sentinel-4 [13], and the Geostationary Environment Monitoring Spectrometer (GEMS) [14,15]. The goal of the constellation project using these satellites is to monitor global air quality such as the long-range transport of aerosols and the emission of pollutants in high spatiotemporal resolution [16]. Among them, GEMS was launched in February 2020 with a planned 10-year lifetime. GEMS observes the reflected radiance from the Earth in the UV–Vis region from 300–500 nm, with a resolution of 0.6 nm and a spectral sampling of 0.2 nm [14,15].

Cloud information, cloud top pressure (CTP), and cloud fraction (CF) can be derived from the observed radiance. GOME, GOME-2, and SCIAMACHY use the oxygen A-band (in the three windows of 758–759, 760–761, and 765–766 nm) to obtain information on clouds very rapidly using the Fast Retrieval Scheme for Clouds from the Oxygen A-band (FRESCO) algorithm [17,18]. Additionally, an algorithm using the oxygen B-band (in the three windows: 685–686, 686.8–687.8, and 690–691 nm) has been developed [19]. Obtaining cloud information using a fast scheme can be supplied in the step of near real time (NRT), which can be used to correct or mask the cloud regions in other retrieval algorithms. Due to the spectral coverage of GEMS (300–500 nm), there is a limitation in the retrieval of cloud information using the strong absorption A- and B-bands of oxygen as they are utilized by the FRESCO algorithm. Therefore, an alternative to FRESCO is necessary to acquire fast cloud information using GEMS spectral information. There are two methods to obtain cloud information without using oxygen bands. The first method is to use the filling-in effect on the Fraunhofer lines by rotational Raman scattering (RRS) at 345 to 354 nm, named OMCLDRR [20], and the other method is the Differential Optical Absorption Spectroscopy (DOAS) method using the absorption of $O_2$–$O_2$, named OMCLDO2 [21,22]. The OMCLDO2 algorithm is based on the assumption of the mixed Lambertian equivalent reflectivity (MLER) for the cloud treatment. In order to derive the CTP and CF, OMCLDO2 was used to perform a DOAS fit between 460 and 490 nm, and the look-up table (LUT) inversion technique was applied from the continuum reflectance at 477 nm and $O_2$–$O_2$ slant column density (SCD). These two methods provide the cloud parameter information such as the operational level 2 cloud products of the OMI. Furthermore, an advanced spectral fitting method based on the $O_2$–$O_2$ absorption band using Geometry-dependent Lambertian Equivalent Reflectivity (GLER), similar to OMCLDO2, was developed. In this new version of the OMCLDO2 algorithm [23], temperature dependence correction of $O_2$–$O_2$ was applied based on DOAS fitting of [21,22]. Additionally, the method of removing outliers from the spectral fitting was implemented, and the number of nodes for surface albedo and surface pressure of LUT were increased.

In this study, we developed a novel fast and robust cloud retrieval algorithm using triplet of wavelengths of the $O_2$–$O_2$ absorption band around 477 nm. The algorithm is based on the absorption difference of the $O_2$–$O_2$ due to the change of photon penetration, which depends on the CTP. This algorithm is similar to the previous algorithms in that it uses the $O_2$–$O_2$ absorption band [21–23], but it differs in that it applies the peak/wing ratio of radiance at three selected wavelengths. A detailed description of the algorithm is presented in Section 3. In Section 4, the retrieval results of the proposed algorithm for the CTP and CF are evaluated in comparison with the results obtained with FRESCO using GOME-2 observation data.

## 2. Instrument and Data

GOME-2 is the nadir-scanning UV–Vis spectrometer onboard the MetOp series (MetOp A, B, and C) of satellites. The MetOp satellites observe the entire Earth in a sun-synchronous

orbit at an altitude of 820 km and overpass the equator at approximately 09:30 local time (LT). The principal purpose of this sensor is to monitor not only the total column amount of ozone, but also the trace gases in the atmosphere around the world for research related to air pollution and climate change. GOME-2 measures the backscattered radiance from the Earth and solar irradiance between 240 and 790 nm in four Main Science Channels (MSC). Moreover, GOME-2 measures the polarization state of the signal backscattered by the atmosphere simultaneously using the Polarization Measurement Device (PMD). Since GOME-2 is a polarization-sensitive instrument, the radiometric polarization calibration of MSCs is performed using measurement information from the PMD. PMD measures a lower spectral resolution than the MSCs, but it has a higher spatial resolution, which is then utilized to determine the sub-pixels of cloud cover for the MSCs [10]. GOME-2 covers the entire Earth within 1.5 days with a swath width of 1920 km. The horizontal spatial resolution is $80 \times 40$ km$^2$ at a nadir-viewing point. The specifications of the GOME-2 instrument are summarized in Table 1. For more details on the instrument characteristics of GOME-2, refer to [10].

**Table 1.** Specifications of the Global Ozone Monitoring Experiment-2 (GOME-2) instrument.

| Parameter | GOME-2/MetOp-B | |
| --- | --- | --- |
| | Main Science Channel (MSC) | Polarization Measurement Device (PMDs) |
| Spectral Range | 239–791 nm | 312–790 nm |
| Spectral Sampling | 0.12–0.21 nm | 0.62–8.8 nm |
| Spectral Resolution | FWHM 0.29–0.55 nm | FWHM 2.9–37 nm |
| Spatial Resolution | $80 \times 40$ km | $10 \times 40$ km |
| Swath Width | 1920 km | |

In this study, we used GOME-2/MetOp-B Level 1B (L1B) channel 3 data covering 395 to 604 nm with a spectral resolution of 0.55 nm for the retrieval of the cloud parameters. In the default scan mode scenario, the GOME-2 scan mirror sweeps from negative (east) to positive (west) viewing angles and then returns to negative viewing angles (see Figure 5 of [10]). Due to this scan pattern, we only used forward scan mode data to avoid overlapping points in the observed data.

The derived cloud information in this study, CTP and CF, was evaluated with the FRESCO+ (hereafter, FRESCO) cloud algorithm from the GOME-2 measurement (FRESCO data are available at http://www.temis.nl/fresco/). In many previous studies on the retrieval of cloud information, the results of FRESCO were validated with other cloud algorithms and ground-based observation, and its accuracy has been verified. For example, [17] compared the cloud parameters from Along-Track Scanning Radiometer 2 (ATSR-2) with GOME FRESCO; [18] compared Atmospheric Radiation Measurement (ARM) active ground-based remote sensing cloud boundaries data and SCIAMACHY FRESCO; and [19] compared GOME-2 FRESCO with the Cloudnet level 2 classification product composed of the vertical Doppler cloud radar and backscatter lidar. They showed good agreement with each comparison target of the FRESCO algorithm.

## 3. Methodology

### 3.1. Retrieval Algorithm

The principal atmospheric components that absorb sunlight in the UV–Vis region are ozone ($O_3$), sulfur dioxide ($SO_2$), nitrogen dioxide ($NO_2$), formaldehyde (HCHO), bromine monoxide (BrO), water vapor ($H_2O$), and aerosols. In addition to these trace gases, absorption occurs broadly due to the collisions between oxygen molecules, which are named oxygen dimers (hereafter, $O_2$–$O_2$) [24,25]. The absorption cross-sections of the oxygen dimer are shown in Figure 1. As can be seen in Figure 1, there are several strong absorption bands centered at 344.0, 360.7, 380.2, 446.7, and 477.0 nm within the 300–500 nm spectral range, the strongest of which occurred at around 477 nm with a full width at half maximum (FWHM) = 5.39 nm (at 293K) [26]. At wavelengths outside of 460 and 490 nm

of the absorption band of 477 nm, the absorption effect of $O_2$–$O_2$ is very small and can be negligible.

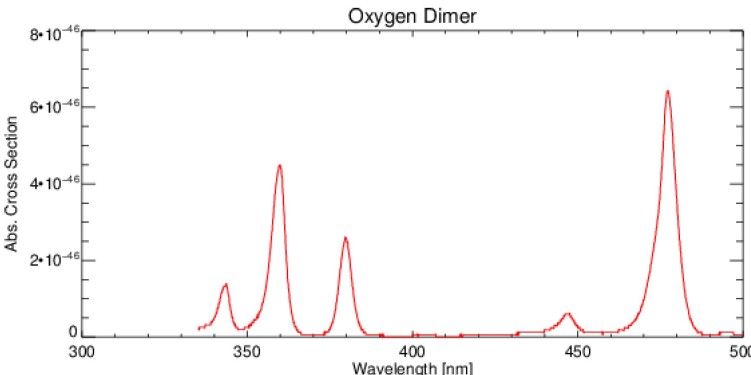

**Figure 1.** Absorption cross-sections of the oxygen dimer at 293 K determined from [27].

The cloud pressure retrieved using the $O_2$–$O_2$ absorption band refers to the optical center of the cloud [15]. A fast cloud retrieval algorithm using a triplet of wavelengths of the $O_2$–$O_2$ band around 477 nm (hereafter, CLDTO4) regards the top surface of the cloud as a Lambertian equivalent reflector and derives the cloud information. CLDTO4 assumes that the influence of aerosols in the atmosphere is very small compared to that of clouds and ignores the effect of aerosols in the retrieval process. When the atmosphere contains pollutants such as dust, biomass burning, or volcanic ash, these aerosols act as absorbing materials. In such cases, CTP misrecognizes a cloudless area as a cloud or underestimates it.

Our new CLDTO4 algorithm is similar to other current cloud retrieval algorithms [21–23] that are based on the $O_2$–$O_2$ absorption band at 477 nm and use the LUT inversion approach to retrieve the cloud parameters. However, CLDTO4 differs in that it uses peak/wing ratio information at only three wavelengths without the need to perform spectral fitting of wide $O_2$–$O_2$ absorption bands or match $O_2$–$O_2$ SCD like the other algorithms from the DOAS method. This reduces the time required for the calculation of the retrieval algorithm.

The CF can be determined using the independent pixel approximation (IPA) method [21,27] as follows:

$$C_f(\lambda) = \frac{I(\lambda) - I_{clr}\left(\lambda, A_{sfc}, P_{sfc}\right)}{I_{cld}(\lambda, A_{cld}, P_{cld}) - I_{clr}\left(\lambda, A_{sfc}, P_{sfc}\right)} \tag{1}$$

where $I$, $A$, and $P$ are the normalized radiance signal (hereafter, NRS) at wavelength ($\lambda$), albedo, and pressure, respectively. $I(\lambda)$ is the observed NRS by the satellite. NRS is determined as TOA radiance divided by solar irradiance. The subscripts *sfc*, *clr*, and *cld* represent the state of the surface, clear, and cloud, respectively. $A_{sfc}$ and $A_{cld}$ denote the albedo of the ground surface and cloud. $P_{sfc}$ and $P_{cld}$ denote the surface pressure and CTP, respectively. The IPA method assumes that in a given pixel observed by a satellite, there exists a cloudless surface and a cloudy region, and they are weighted by the corresponding radiance. With the consideration of the plane-parallel atmosphere, it is assumed that there is no transport of photons in the horizontal direction, and radiative transfer is only in the vertical direction [27]. In other words, the CF can be determined as the ratio of the measured radiance at the satellite and the theoretically calculated cloud radiance by RTM, which removes the clear sky surface contribution.

CLDTO4 uses the following three-wavelength positions to determine the cloud effect: 477 nm, which has the strongest absorption effect of $O_2$–$O_2$; and 469 and 485 nm on either side of the $O_2$–$O_2$ absorption band, which are hardly affected by the $O_2$–$O_2$ absorption by clouds. Hereafter, 477 nm is named 'peak=core', and 469 and 485 nm are named 'wings'. The amount of absorption by $O_2$–$O_2$ changes depending on the length of the

optical path through which sunlight passes. Consequently, the presence of clouds can lead to a difference in radiances between the peak and wings. The average of both wings ($I_{wing}$) and the ratio of the peak to the wings ($R$) are defined as follows:

$$I_{wing} = (I_{469} + I_{485}) / 2 \tag{2}$$

$$R = I_{477} / I_{wing} \tag{3}$$

where $I_{469}$, $I_{477}$, and $I_{485}$ are the normalized radiance values at 469, 477, and 485 nm, respectively. Figure 2 shows that the change of $R$ depends on the average of the wing radiance for various geometric angles. These values were calculated using the radiative transfer model, which is described in the next section. The sensitivity test for the variation of solar zenith angle (SZA) and viewing zenith angle (VZA) was performed. The RAA and total column amount of ozone were fixed as 90° and 325 Dobson units (DU), respectively. The SZA and VZA varied from 15° to 40°, and 75°. The average wing radiance represents the reflectivity (solid lines in Figure 2) of the observed pixel from the satellite, and $R$ refers to the influence of the CTP (dotted lines in Figure 2). Each line in Figure 2 corresponds to the values of the nodes presented in Table 2. From Figure 2, it can be noticed that $R$ is less sensitive to changes in reflectivity and changes linearly with changes in surface pressure. As the altitude at which sunlight is reflected by the cloud decreases (and thus as the surface pressure of the cloud increases), $R$ decreases. $R$ becomes close to 1 when the amount of air molecules over the cloud decreases as the altitude of the cloud is high, approaching the tropopause. In addition, as the SZA increases, the slant optical path through which sunlight passes increases. Therefore, the fluctuation width of $R$ is larger than those of lower SZA. At low surface reflectivity (<0.1), changes due to surface pressure are sharp. This can become a source of error for low-elevation clouds or for pixels with low fractional cloud cover.

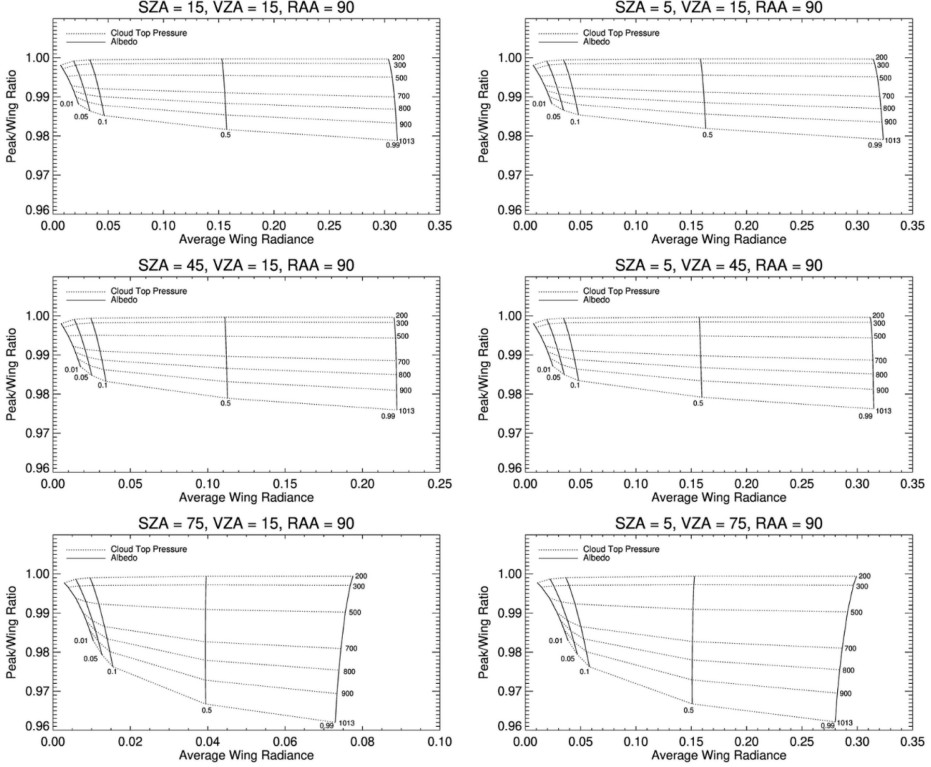

**Figure 2.** Change of peak/wing ratio according to average wing radiance for various geometric angles. The solid lines indicate a surface albedo of 0.01, 0.05, 0.1, 0.5, and 0.99 (from left to right), and the dotted lines indicate the cloud top pressure (CTP) of 1013, 900, 800, 700, 500, 300, and 200 hPa (from bottom to top), respectively.

**Table 2.** Summary of parameters and nodes used to construct the look-up table (LUT).

| Parameter (Unit) | Nodes |
|---|---|
| Wavelength (nm) | 469, 477, 485 |
| SZA (°) | 0.1, 15, 30, 45, 60, 75, 85.9 |
| VZA (°) | 0.1, 15, 30, 45, 60, 75, 85.9 |
| RAA (°) | 0.1, 30, 60, 90, 120, 150, 179.9 |
| Surface albedo | 0.01, 0.05, 0.1, 0.5, 0.99 |
| Surface pressure (hPa) | 1013, 900, 800, 700, 500, 300, 200 |
| Ozone profiles (DU) | L225, L275, L325, L375, L425, L475, M175, M225, M275, M325, M375, M425, M475, M525, M575 |

L and M indicate low-latitude (<30°) and mid-latitude (>30°).

### 3.2. Description of Look-Up Table (LUT)

The LUT method is an effective way to save computation time when simulating the satellite-observed radiance, as it takes a long time to calculate using the radiative transfer model in real-time. For this purpose, the LUT was prepared for several parameters affecting radiance. Radiance was pre-calculated using Vectorized Linearized Discrete Ordinate Radiative Transfer (VLIDORT; [28,29]) as a function of SZA, VZA, RAA, surface albedo, surface pressure, and ozone profiles. The atmospheric condition is assumed to be a molecular atmosphere without aerosols. The radiative transfer model (RTM) simulation was executed in vector mode to account for the polarization effect. This is because the failure to consider polarization in the simulation of scalar mode can cause an error of up to 10% at the TOA [30,31]. The atmospheric profiles (i.e., temperature, humidity, and gases) were adopted from the Air Force Geophysics Laboratory (AFGL) atmospheric constituent databases for the United States standard atmosphere 1976 (US76; [32]). The calculation included absorption by $O_3$, $NO_2$, and $O_2$–$O_2$. For $NO_2$, a fixed profile (total amount of $NO_2$ was 0.215 DU) was applied to the RTM simulation. The ozone profiles were constructed based on Version 8 (V8) of the Total Ozone Mapping Spectrometer (TOMS) climatology data [5]. These profiles were classified into low-latitude (L) and mid-latitude (M) depending on the location of the latitude with regard to the total column amount of ozone. The threshold for distinguishing between mid and low latitudes was 30° south/north latitude. The LUT contains seven nodes of SZA, seven nodes of VZA, and seven nodes of RAA. The angles of step size of SZA, VZA, and RAA were 15°, 15°, and 30°, respectively. Surface albedo and surface pressure were calculated for five and seven nodes, respectively. Two-thirds of the Earth is covered by sea, which has low reflectivity in visible regions [33,34]. The surface at high-latitude regions or high mountainous areas covered with snow or ice has high reflectivity. This means that most regions of the Earth, except the permafrost and polar regions, have relatively low reflectivity. Therefore, the step size of the LUT is constructed smaller increments at relatively low reflectivity values (less 0.1) and more sparsely at higher values. The parameter nodes are summarized in Table 2.

## 4. Results

### 4.1. CLDTO4 Retrievals from GOME-2 Observation

We applied CLDTO4 to GOME-2 data to evaluate the performance. The cloud parameters were retrieved by CLDTO4 on 20 September 2016, a randomly selected date. The variables (i.e., total column amount of ozone, surface pressure, and surface reflectivity), except for the radiance and geometry of the satellite used in CLDTO4, were adopted by climatology data.

The observed TOA radiance of each pixel consisted of the clear sky sub-pixel and the CF weighted cloud sub-pixel as described in Equation (1). In order to know the contribution of the clear sky, we require the information of $A_{sfc}$ and $P_{sfc}$. We adopted the GOME-2 Lambertian Equivalent Reflectivity (LER) database [35] with a resolution of 0.25° × 0.25° to consider the temporal and spatial variation of the surface reflectivity. The surface reflectivity was interpolated to 477 nm and applied to $A_{sfc}$. We used the Earth

TOPOgraphy (ETOPO)-2 dataset [36] to obtain the terrain height information. The ETOPO-2 dataset was generated from digital databases of seafloor and land elevations on a 2-minute latitude/longitude grid. The coverage of ETOPO-2 is 90°S to 90°N in latitude, and 180°W to 180°E in longitude. In order to consider only the surface altitude, the submarine region with a negative value was replaced with zero. The terrain height was converted to $P_{sfc}$ using the following barometric formula.

$$P_{sfc} = P_0 exp^{(-z/H)}$$

(4)

where $P_0$ is the sea level pressure (1013.25 hPa); $z$ is the altitude at the level; and $H$ is the scale height. The scale height was assumed to be 8 km. $R$ and $I_{wing}$ were calculated using Equations (2) and (3), respectively, from the radiance of the three selected wavelengths (469, 477, and 485 nm) observed in each individual pixel of GOME-2. The pre-constructed LUT was multi-dimensionally interpolated to the values of input variables (SZA, VZA, RAA, and total column amount of ozone) given at each pixel. At this time, the total column amount of ozone was adopted from the McPeters and Labow (ML) climatology data [37] (seasonal ozone profile merged data composed of Aura Microwave Limb Sounder (MLS) and ozone sonde data at intervals of 10° from −90°S to + 90°N). The total column amount of ozone was applied by integrating the amount of ozone in each vertical layer. $P_{cld}$ and $C_f$ could be obtained from the optimized values that satisfied the calculated $R$ and $I_{wing}$ from the LUT. The retrieved CTP and CF by CLDTO4 are illustrated in Figure 3a,b.

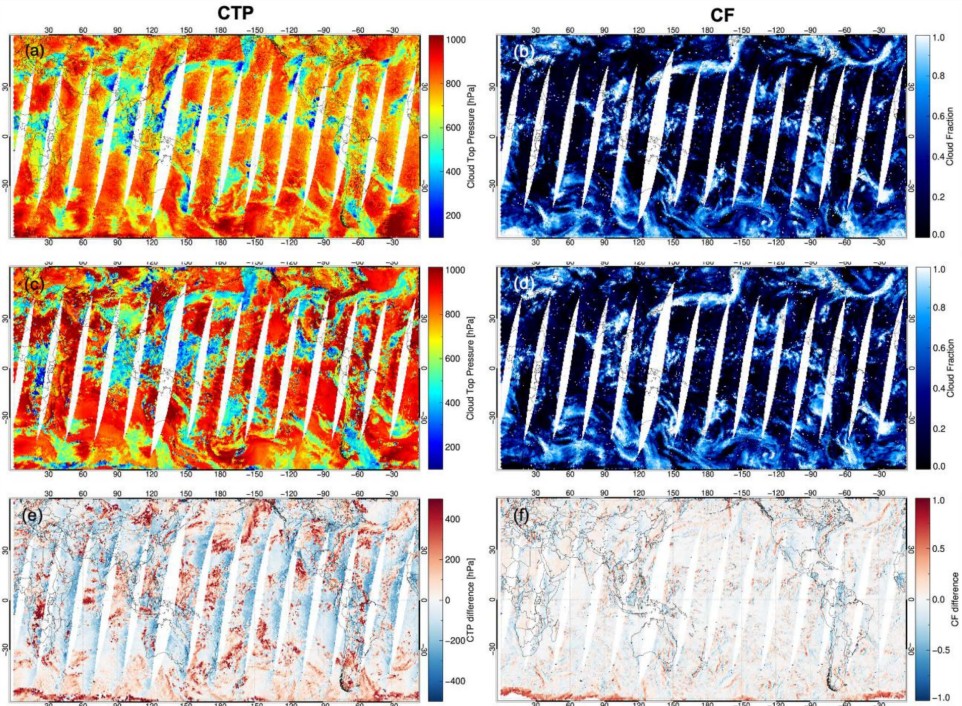

**Figure 3.** Retrieval result of CTP and cloud fraction (CF) by (**a**,**b**) fast cloud retrieval algorithm using a triplet of wavelengths of the $O_2$–$O_2$ band around 477 nm (CLDTO4), (**c**,**d**) Global Ozone Monitoring Experiment-2 (GOME-2) Fast Retrieval Scheme for Clouds from the Oxygen A band (FRESCO), and (**e**,**f**) difference between CLDTO4 and FRESCO on 20 September 2016.

*4.2. Inter-Comparison of Cloud Parameters with Fast Retrieval Scheme for Clouds from the Oxygen A Band (FRESCO)*

The CTP and CF were compared with the values retrieved from GOME-2 FRESCO to verify the performance of CLDTO4. As mentioned earlier, FRESCO applies a nonlinear fitting method based on the $O_2$-A absorption band at 760 nm, and not a DOAS method such as OMI. An additional difference between CLDTO4 and FRESCO is that the wavelength

applied by approximating the surface reflectance data is 760 nm for FRESCO and 477 nm for CLDTO4. The surface reflectivity in the visible region is higher over desert and evergreen forest than in the ultraviolet region and lower over ocean [34,35,38–40]. FRESCO, which uses the visible region, has a high sensitivity to the variation of radiance with the reflectance by surface type. Since GOME-2 FRESCO is also a result of the retrieval algorithm rather than an actual in situ observation, it is difficult to consider as a true value for cloud information. Nevertheless, the mutual comparison of these two algorithms is meaningful for the purpose of analyzing the difference between the $O_2$–$O_2$ method of CLDTO4 and the $O_2$-A method of FRESCO in the retrieval of the cloud parameters and to evaluate the performance of CLDTO4.

Figure 3 shows the retrieved results for CTP and CF from both algorithms on 20 September 2016. The CTP results retrieved from CLDTO4 and FRESCO are shown in Figure 3a,c, respectively; CFs are shown in Figure 3b,d, respectively. The differences between the two retrieval algorithms (CLDTO4–FRESCO) are illustrated in Figure 3e,f. Overall, the qualitative behavior of both algorithms detecting cloud regions was very similar. The negative differences between the two algorithms occurred in the region where the VZA of GOME-2 was large, at the right edge of the cross-track (see Figure 3e). The CTP of FRESCO was relatively low on the right side of the along-track direction of GOME-2. The mean value of the negative difference that occurred on the right side of the along-track was $-115 \pm 87.87$ hPa. This is noticeable where the derived CTP is close to 1 atm (1013.25 hPa). CLDTO4 tends to treat the CTP slightly lower for clear sky or low-altitude clouds. This is because the low radiance intensity is very sensitive to the variation of CTP, as mentioned in the previous section. However, these differences can be ignored because most of these areas correspond to clear sky, which has a very low CF (<0.2). CLDTO4 overestimates the CTP in the tropical and sub-tropical regions with respect to FRESCO (see Figure 3c). This means that when the fraction of clouds occupied by pixels is small in tropical or sub-tropical regions, FRESCO is retrieved as a relatively high cloud compared to CLDTO4. Figure 4 presents CTP and CF as probability distribution functions (PDFs) for CLDTO4 and FRESCO. The CTP distribution peak of CLDTO4 occurred at around 800 hPa, whereas that of FRESCO occurred at 850 hPa. The center of the distribution of FRESCO was skewed toward lower cloud pressure with respect to CLDTO4, which means that CLDTO4 is considered a rather lower pressure for low-altitude cloud compared to FRESCO. In contrast, FRESCO sees a greater proportion of CTP than CLDTO4 at above 600 hPa. In pixels where clouds exist (CF > 0.1), FRESCO treats clouds at a slightly higher percentage than CLDTO4. In contrast, CLDTO4 has a high proportion at almost clear sky (CF < 0.1). Figure 5 shows a boxplot of the distribution of CTP along the range of CF in each algorithm. CF was divided from 0 to 1 with 0.1 intervals. When CF was between 0 and 0.1 (i.e., an almost clear sky pixel), the maximum cloud height was 550 hPa for CLDTO4, compared to 100 hPa for FRESCO. In CLDTO4, the range of cloud height steadily increased as CF increased, while FRESCO obtained high-altitude clouds (close to 200 hPa) in all CF ranges. This means that FRESCO treats the cloud as higher for the optically thin fractional cloud or transparent cloud. Meanwhile, for the section of CF from 0–0.1 to 0.9–10, the median CTP of CLDTO4 gradually decreased, with small fluctuations from 789.2 to 709.6 hPa. On the other hand, FRESCO decreased, dramatically from 803.6 to 642.3 hPa, as CF increased. This indicates that FRESCO considers clouds higher-altitude clouds when the percentage of clouds within the pixel is high. CF showed good agreement, as the difference between the two algorithms was very small. Unlike CTP, there was no bias characteristic due to the observation geometry.

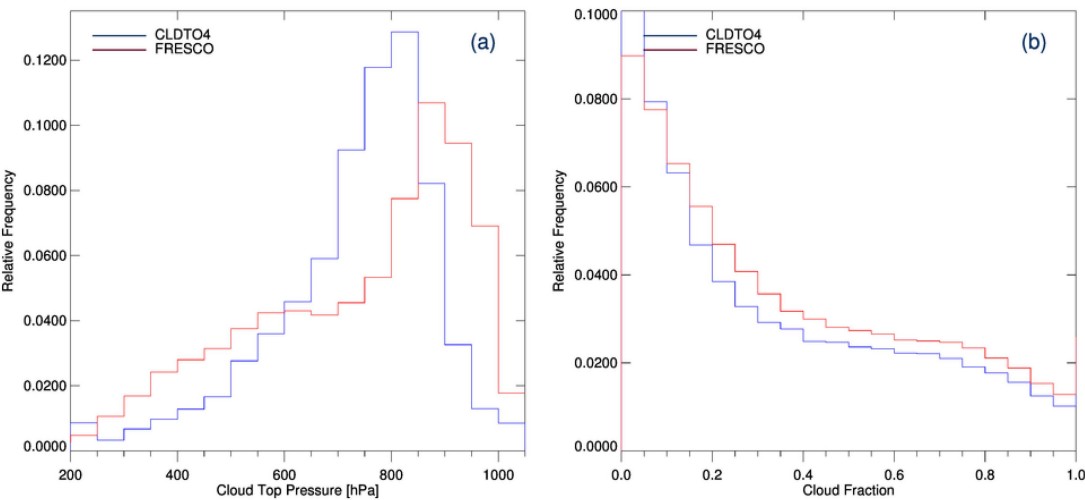

**Figure 4.** Probability distribution of (**a**) CTP and (**b**) CF by CLDTO4 and FRESCO.

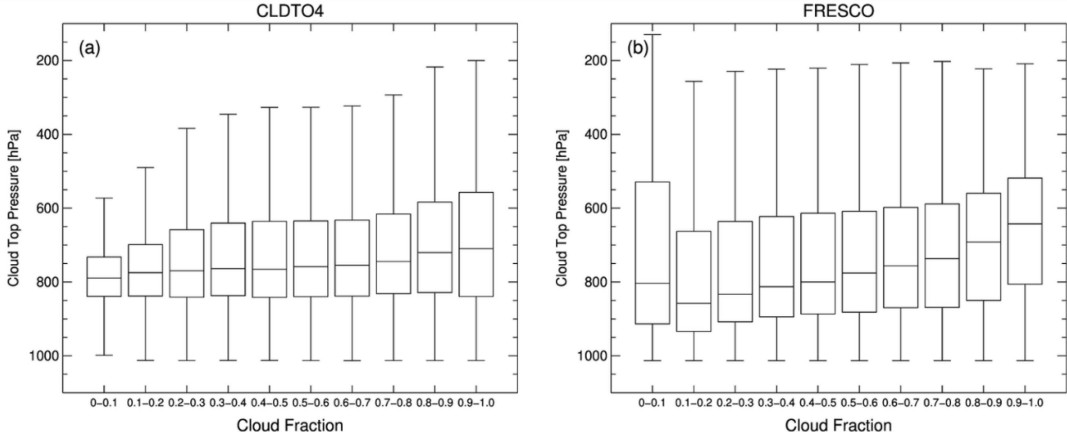

**Figure 5.** Boxplot of distribution of CTP as function of CF for each cloud retrieval algorithm: (**a**) CLDTO4 and (**b**) FRESCO.

Figure 6 shows an example of the cross-section of CTP and CF as a function of latitude for each calculation algorithm. The variation of latitude indicates the location of satellite pixels along the GOME-2 flight direction. Although there were some discrepancies, the tendency of the CTP and CF of both algorithms for the cloud region was very similar. As above-mentioned, between the latitude of 20°S to 20°N (tropical and sub-tropical regions), there were features in which CLDTO4 tended to assign the CTP as higher than FRESCO for low clouds and slightly lower for high clouds. FRESCO detects the clouds as high altitude at the edges of several consecutive cloud distributions. In this GOME-2 track line, the mean differences (CLDTO4–FRESCO) of CTP and CF between CLDTO4 and FRESCO were $29 \pm 163.59$ hPa and $0.19 \pm 0.67$, respectively.

Figure 7 presents the comparison of the results between CLDTO4 and FRESCO for CTP and CF as a histogram of bias and a density scatterplot. These are the results for all of the outputs from the GOME-2 observations shown in Figure 3. As mentioned earlier, the CTP retrieved by CLDTO4 tended to be underestimated in the clear region (CF < 0.2) compared to FRESCO. In particular, the difference wase up to 100 hPa for low clouds, where FRESCO's CTP is 800–900 hPa (Figure 7b). The retrieved values of CF obtained using CLDTO4 were very similar to those retrieved using FRESCO, except in some high-latitude regions with bright surfaces. This is because a ground surface with high reflectivity (e.g., snow or ice) can be mistaken for clouds. Both algorithm results for CF almost followed the 1:1 line, although the CF of FRESCO was slightly large. The biases in CTP and CF had a Gaussian distribution. The statistical results for CTP and CF are summarized in Table 3. For

all sky, in the case of CTP, the correlation coefficient, bias, root mean square error (RMSE), and mean absolute error (MAE) were 0.74, −3.56, 77.5, and 34.75, respectively. For CF, the correlation coefficient, bias, RMSE, and MAE were 0.67, 0.11, 0.11, and 0.05, respectively. Furthermore, where the presence of clouds was significant (CF > 50%), the two algorithms showed better agreement. The correlation coefficient, bias, RMSE, and MAE for CTP were 0.79, −2.32, 48.61, and 15.56, respectively. This indicates that the CLDTO4 method provides cloud information with a high level of accuracy that matches that of FRESCO.

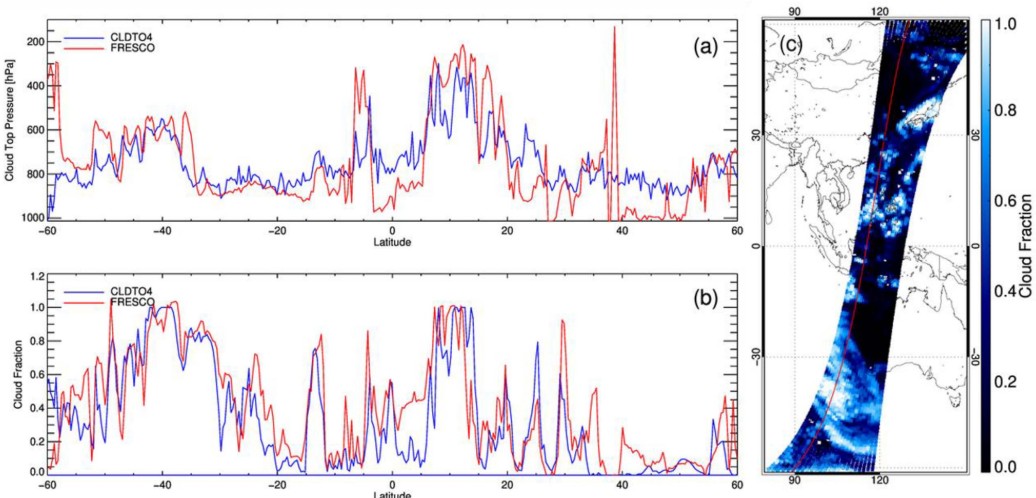

**Figure 6.** Intercomparison of (**a**) CTP and (**b**) CF was retrieved from CLDTO4 and FRESCO. The cross-section is a GOME-2 orbit as a function of the latitude from 0108 to 0250 UTC on 20 September 2016. (**c**) The trajectory of the over-passed track of GOME-2. The track line used in (**a**,**b**) is represented as a red line. The overlapped background image is the CF of CLDTO4.

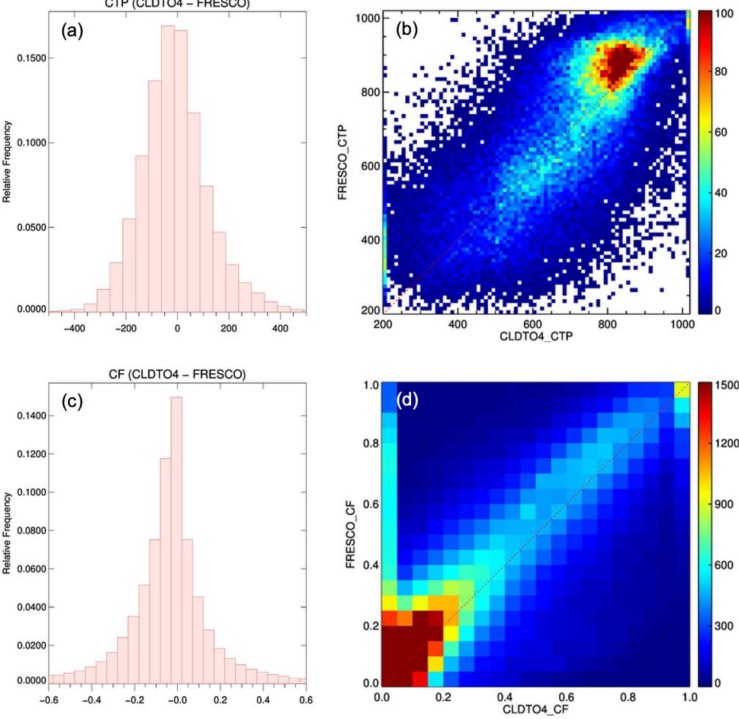

**Figure 7.** (**a**) Histogram of bias and (**b**) density plot for CTP. (**c**) Histogram of bias and (**d**) density plot for CF.

**Table 3.** Statistical results of CTP and CF for coverage of cloud pixels in (**a**) all sky and (**b**) significant CF (>0.5).

| Parameter | (a) All Sky | | (b) CF > 0.5 | |
|---|---|---|---|---|
| | CTP (hPa) | CF | CTP (hPa) | CF |
| Correlation Coefficient | 0.74 | 0.67 | 0.79 | 0.61 |
| Bias | −3.56 | 0.11 | −2.32 | 0.004 |
| RMSE | 77.58 | 0.11 | 48.61 | 0.06 |
| MAE | 34.75 | 0.05 | 15.56 | 0.01 |

## 5. Discussion

CLDTO4 only uses a triplet of wavelengths, unlike the spectral fitting method, which uses the whole set of wavelengths in the absorption band to derive the cloud information in a short time. Therefore, CLDTO4 is sensitive to the radiance spectral feature affected by the radiometric calibration quality and the polarization sensitivity of the instrument. This is related to the importance of securing radiometric accuracy in the L0–1 process.

CLDTO4 was developed for GEMS, but in this study, the evaluation was performed using GOME-2 data. When cloud parameters are derived by applying CLDTO4 to GEMS, cloud structure information on a smaller scale can be provided because the spatial resolution of GEMS is smaller than that of GOME-2. Both satellites observed the radiance within the wavelength range including the $O_2$–$O_2$ absorption band for utilizing the CLDTO4, but there were differences in spectral resolution, slit function, and signal-to-noise ratio (SNR), which could affect the shape of the radiance spectrum. This can lead to differences in cloud retrieval results. Furthermore, GOME-2 and GEMS are LEO and GEO, respectively. In general, the range of VZA of GEO is relatively larger than that of LEO. Since the calculation is not accurate for large VZA and SZA in the radiation transfer process, assuming a plane-parallel atmosphere, the uncertainty of the cloud retrieval result can increase in the region with large VZA of GEMS.

In this study, the cloud parameters of CLDTO4 achieved good performance as a result of comparison with FRESCO, but there remains scope for improvement in the future. For example, CLDTO4 underestimated CTP compared to FRESCO in low cloud states. In addition, one fixed $NO_2$ profile was used in the LUT configuration. In regions where the emissions of $NO_2$ are high, this can be a source of error in cloud retrieval. Therefore, additional correction for the amount of $NO_2$ is needed. We constructed the LUT by assuming the Rayleigh atmosphere, which does not consider the aerosol effect. This assumption treats the amount of radiation absorbed from the atmosphere as larger for turbid atmospheres containing pollutants such as mineral dust, biomass, and volcanic ash. Then, CTP can misrecognize a cloudless area as a cloud or underestimate it. The absorption cross-section of $O_2$–$O_2$ has a temperature dependence [21,23,41]. CLDTO4 does not consider the variation of the atmospheric temperature profile to reduce temperature uncertainty. Moreover, the absorption cross-sections exist not only for $O_2$, $NO_2$, and $O_2$–$O_2$, but also for water vapor within the spectral range of GEMS, so there is potential for improving the LUT by considering water vapor.

In the future, we will apply the actual slit function of GEMS and optimize the LUT configuration to improve the proposed algorithm's accuracy, because the LUT nodes at regular intervals may cause nonlinear interpolation errors. Furthermore, we will retrieve cloud information from the measured data obtained since the GEMS was launched as well as verify the CLDTO4 with the GEMS operational cloud algorithm and cloud radar or lidar (ground-based or satellite) and continue to improve its precision.

## 6. Summary and Conclusions

Clouds play an important role in the amount of sunlight reflected out of the atmosphere. In order to produce trace gas or aerosol information with high accuracy from the

radiance that is observed from satellites at the TOA in determining NRT, it is necessary to obtain accurate cloud information quickly. The fundamental purpose of this study was to retrieve the cloud information in a short time within the spectral region of GEMS. CLDTO4 used the absorption difference at 469, 477, and 485 nm concerning the optical path change depending on the presence of clouds. To reduce the calculation time, a LUT was constructed for various conditions considering the polarization effect using VLIDORT.

The results of CTP and CF retrieved by CLDTO4, developed in this study, were compared with the GOME-2 FRESCO algorithm results. There was a small discrepancy in the proposed algorithm, which underestimated the CTP of low cloud (up to 100 hPa difference) with respect to FRESCO, but the results for CTP and CF were very similar and showed good performance. In the regions with significant CF (>0.5), the correlation coefficient was 0.79, which was 0.05 higher than the all sky condition (0.74), and the RMSE was 48.61 hPa, which was 28.97 hPa lower than the all sky condition (77.58 hPa). When the CF was between 0 and 0.1, which means almost clear sky, FRESCO tended to yield the clouds as higher-altitude than CLDTO4. The CLDTO4 method using the $O_2$–$O_2$ band at 477 nm was less sensitive to fractional clouds (CF < 0.2) and transparent clouds at high altitude than FRESCO. The penetration length of photons of reflected sunlight differed in two channels with different absorption characteristics such as the $O_2$–$O_2$ band and the $O_2$-A band. Therefore, the method of using each independent absorption band can cause differences in the retrieval results for CTP. The result of CF of CLDTO4 is suitable for determination of the threshold value applied to the cloud masking of the pixel for the retrieval algorithms of trace gases such as $NO_2$, $SO_2$, and HCHO. The slight difference of CTP that occurred in low altitude clouds compared to FRESCO could have a significant effect on correction for the amount of ozone under the cloud layer to retrieve the total amount of ozone in the cloud region. For example, if the CTP is lower than the actual value, it can be assumed that there is a greater amount of ozone under the cloud in that pixel. Then, the total ozone amount will be overestimated. Nevertheless, the cloud parameters obtained using the proposed algorithm were comparable to those obtained using FRESCO. Therefore, it is sufficient to be utilized for the retrieval of trace gases and aerosol properties.

**Author Contributions:** This work was made possible by significant contributions from all authors. Conceptualization, X.L. and H.C.; Methodology, H.C., X.L., and K.-M.L.; Software, H.C. and J.S.; Validation, H.C. and K.-M.L.; Formal analysis, H.C. and K.-M.L.; Investigation, H.C. and X.L.; Writing—original draft preparation, H.C.; Writing—review and editing, K.-M.L., X.L., G.G.A., and J.K.; Visualization, H.C.; Supervision, K.-M.L.; Funding acquisition, K.-M.L. and J.K. All authors have read and agreed to the published version of the manuscript.

**Funding:** This research was supported by the Korea Ministry of Environment (MOE) through the "Public Technology Program based on Environmental Policy (2017000160002)".

**Institutional Review Board Statement:** Not applicable for studies not involving humans or animals.

**Informed Consent Statement:** Not applicable for studies not involving humans.

**Data Availability Statement:** The data that support the findings of this study are available from the corresponding author upon reasonable request.

**Acknowledgments:** The authors would like to express their gratitude to the reviewers for their valuable comments and suggestions for improving this manuscript.

**Conflicts of Interest:** The authors declare no conflict of interest.

## Abbreviations

| | |
|---|---|
| AFGL | Air Force Geophysics Laboratory |
| ATSR | Along-Track Scanning Radiometer |
| ARM | Atmospheric Radiation Measurement |
| CF | Cloud Fraction |
| CLDTO4 | fast CLouD algorithm using the Triplet of wavelengths around Oxygen-dimer |
| CTP | Cloud Top Pressure |
| DU | Dobson Unit |
| DOAS | Differential Optical Absorption Spectroscopy |
| ETOPO | Earth TOPOgraphy |
| FRESCO | Fast Retrieval Scheme for Clouds from the Oxygen A band |
| FWHM | Full Width at Half Maximum |
| GEMS | Geostationary Environment Monitoring Spectrometer |
| GEO | Geostationary Earth Orbit |
| GLER | Geometry-dependent Lambertian Equivalent Reflectivity |
| GOME-2 | Global Ozone Monitoring Experiment-2 |
| IPA | Independent Pixel Approximation |
| LEO | Low Earth Orbit |
| LT | Local Time |
| LUT | Look Up Table |
| MAE | Mean Absolute Error |
| MLER | Mixed Lambertian Equivalent Reflectivity |
| MLS | Microwave Limb Sounder |
| MSC | Main Science Channels |
| NRS | Normalized Radiance Signal |
| NRT | Near Real Time |
| $O_2$-$O_2$ | Oxygen Dimer |
| PMD | Polarization Measurement Device |
| RAA | Relative Azimuth Angle |
| RMSE | Root Mean Square Error |
| SZA | Solar Zenith Angle |
| VLIDORT | Vectorized Linearized Discrete Ordinate Radiative Transfer model |
| VZA | Viewing Zenith Angle |

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
