# Peer review of "A Fast Retrieval of Cloud Parameters Using a Triplet of Wavelengths of Oxygen Dimer Band around 477 nm"

_remotesensing, doi:10.3390/rs13010152_

Round 1
Reviewer 1 Report
The paper presents a new algorithm to retrieve cloud top pressure and cloud fraction, CLDTO4. The method uses 477.0 nm absorption line of oxygen dimer. Results of CLDTO4 are compared with FRESCO+ results for GOME measurements carried out on 20 September, 2016. The paper is predominantly clear and comprehensible for readers, a series of errors (mainly, minor) and comments are indicated in the attached file. Part of the proposed corrections is highlighted in the file by green color.

Reviewer 2 Report
Choi et al. described a new cloud retrieval algorithm using the O2-O2 absorption band around 477 nm. They use the mean of the wings and ratio of the peak to wing from triplet wavelength at 469, 477 and 485 nm to derive cloud fraction and cloud top height. The authors described the look-up table, then compared the results for GOME-2B data with FRESCO cloud products. The algorithm could be faster than the O2-O2 algorithms which use the full absorption band, also because they do not have to fit the O2-O2 slant columns. It is an interesting paper and have potentials for GEMS.
I think the authors should add a subsection about the retrieval of the GOME-2B cloud product using CLDTO4. Now after the description of the LUT, it is the results. It is not clear how the GOME-2B data is used, how to derive the cloud fraction and cloud pressure from the LUT.
Please give some answers about the specific questions in the paper.
Specific questions
In the retrieval, how are the radiances selected at the 3 wavelengths? Because the GOME-2 data may not have the wavelengths exactly at 469.0, 447.0, 485.0 nm.
The LUT includes ozone, where do you get the ozone data in the retrieval? Where do you get surface pressure in the retrieval?
The surface temperature and/or temperature profile may be also important for the O2-O2 absorptions. Have you tried to correct it?
Eq. 3. Do you need a cloud albedo to use Eq. 3?
Lines 152-154: Do you have corrections for the plane-parallel atmosphere at large solar zenith angle and viewing zenith angle? The calculations are not accurate for plane-parallel atmosphere in these cases.
Line 175, ‘The calculation includes absorption by O3, NO2, SO2, HCHO, and O2-O2.’
How much NO2, SO2, HCHO are included in the calculation? In the LUT, it seems no NO2, SO2, HCHO. Do you used fixed values for NO2, SO2, HCHO? Are there any SO2, HCHO absorptions close to 477 nm? Have you checked if there are any water vapor absorption?
Lines 212-213
“ The negative differences between the two algorithms occur in the region where the viewing zenith angle of GOME-2 is large, at the right edge of the cross-track “
Do you have any explanation about this difference? How large is it? It seems the cloud fraction does not have this feature.
In the caption of Figure 2, you use the ‘…surface pressure of 1013, 900, … hPa’. I think you mean cloud pressure because the cloud is used as a Lambertian surface.
Reviewer 3 Report
Writing in English is a challenge for people for whom it is a second language. The standard of English in the paper will require significant effort to make it a quality scientific document. However, I have indicated in the document using sticky notes where improvements could be made and in some cases proposed alterations. The suggestions are numerous and one hopes these suggestions will be implemented to improve clarity etc.
On the scientific side I did not find any serious deficiencies in the document. What the authors were endeavouring to achieve was clear as was their approach. The data sets assembled were adequate. The mathematical analysis of data sets and the interpretation were probably the strongest part of the paper. It seemed that they were at ease with this task.
While authors indicated that they has read the manuscript prior to submission I suspect that the effort was not always thorough. For example there were errors /typos in the references that should have been detected - English grammar to one side.
I did take exception to one issue in the paper and recommended that the authors develop a workaround ie circumvent the difficulty in an efficient way. The matter related to identifying a physical variable as a radiometric quantity when, in fact, it clearly wasn't. Its units were inconsistent with the units of the quantity in question. In brief, if one divides a radiance by an irradiance the resultant scaled quantity ("normalised" in this case) cannot be a radiance and should not be described in the paper as a radiance. As I stated this ought be rectified by the authors.
As mentioned the edits / suggestions are in sticky note form.

Reviewer 4 Report
GENERAL COMMENTS
Choi et al. present a method to derive cloud properties (top pressure and fraction) from spaceborne radiometers using the absorption of solar radiation by the oxygen dimer. The algorithm is intended to be used on GEMS, but is here demonstrated on GOME-2 images recorded on 20 September, 2016. The topic is of interest for the research community and is suitable for this journal. However, before publication, the language must be completely revised (even taking advantage of an English proofreading service) and the authors should clarify some points still unclear that are raised here below.
SPECIFIC COMMENTS
- Why was 20 September 2016 chosen for demonstrating the operation of the algorithm? Have the authors used the algorithm on another day?
- This new algorithm is said to be "similar to the previous algorithms ... [using] the O2-O2 absorption band [21-24], but differs in that it applies the ratio of radiance intensity" (l. 70). If I'm right, this is the only reference to the novelty of this algorithm compared to the existing ones. I believe that this point should be better discussed, i.e. algorithms in references 21-24 should be introduced first, then differences and advantages of the present method should be discussed in more depth (e.g., in Sect. 3.1). Without this information, I cannot assess the novelty of the present study;
- If I understood it correctly, the algorithm first derives cloud top pressure and albedo based on LUTs (such as the ones in Fig. 2) and peak/wing ratio and the average wind radiance. Afterwards, these results are used to retrieve the cloud fraction based on Eq. 3, which assumes that P_cld and A_cld were retrieved accurately in the first step. However, I presume that the average wing radiance and the peak/wing ratio measured from the satelite are also influenced by the cloud fraction, thus making the retrieval of P_cld and A_cld dependent on the cloud fraction. I do not have understood how the authors have approached this issue, or have I missed some point in the discussion?
- The behavior of CLDTO4 and FRESCO is judged to be "very similar" (l. 212). However, Fig 3e indeed shows some pronounced systematic differences. Please, specify how large are the expected differences / uncertainty;
- The algorithm is intended to be used for the Geostationary Environment Monitoring Spectrometer (GEMS), but is actually demostrated here using GOME-2. The authors should discuss if differences in spectral resolution, spatial resolution, viewing angles (geostationary satellite vs polar satellite), etc. are expected (and how) to influence the performances of the algorithm when applied to GEMS;
- Since FRESCO is here implicitely used as a reference, or at least as a more tested algorithm, it would be useful to provide some information on how it was "validated" or compared to other cloud properties sources;
- It is said that the CLDTO4 algorithm "assumes that the influence of aerosols in the atmosphere is very small compared to that of clouds" (l. 118). Please, discuss in what conditions (e.g., mineral dust, biomass burning, volcanic ash, etc.) the algorithm is expected not to behave well;
- In the conclusions, it is said that "the accuracy of the products obtained using the proposed algorithm is high enough ...". According to what I have understood from the manuscript, this was not proven here. The paper only shows that the products of the new algorithm are comparable to FRESCO;
- Is specular reflection (e.g., sunglint) expected to affect the retrievals?
TECHNICAL REMARKS
I will only list here the most important corrections to the English language needed in the manuscript, since I assume that the text will be thoroughly corrected by a native-speaking colleage or by a professional proofreading service.
- Title and text: the term "triplet wavelengths" sounds wrong to me. I would say "triplet of wavelengths", "three wavelengths" or just avoid id (e.g., "using the oxygen dimer band" in the title);
- l. 15-16: not clear that the solar radiation backscattered by the atmosphere to space is addressed here. Wouldn't it be simpler to just write that clouds change the amount of sunlight reflected to space?
- l. 17-18: not clear what "securing accuracy" means. Do you mean that clouds perturb the retrieval? Same in l. 36;
- l. 20 and l. 28: not clear here what the relation is between GEMS and GOME-2;
- l. 53-54: not clear, rephrase this sentence;
- l. 59-61: please indicate oxygen A- and B-band wavelengths;
- Table 1: it is not clear to me how the information from the two channels is merged in the algorithm;
- l. 103-104: water vapor and aerosols should be also mentioned here;
- Figure 1: please, specify if the cross sections represented here are from reference 27;
- l. 129 and caption of Fig. 2: rather than "various geometric angles" I only see various "solar zenith angles" here;
- l. 154: define "LER";
- Figure 2: please add labels in the plot, indicating the albedo values and the pressure levels for each contour;
- Sect. 3.2: why do you refer to the LUTs values as "nodes"?
- l. 217-219: if CLDTO4 is compared to FRESCO, then the sentence should be written the other way round, i.e. how CLDTO4 compares to FRESCO;
- l. 223-224, "low elevation clouds higher cloud pressure": typo?
- l. 229-230: discuss separately, and more clearly, 1/ the differences in the range; 2/ the differences in the median values;
- Figure 6: it would be nice to see the trajectory (in Fig. 3) of the the track the data were taken from;
- l. 264: do not use "R" for the correlation coefficient. This letter was already used for the radiance ratio;
- l. 267, "the statistical accuracy": not clear what is meant here. Do you mean that the comparison is "better"?
- l. 280-284: this should have been explained in the introduction, not here;
- l. 290-300: what differences are expected?
- correct bibliographic reference n. 16.
Round 2
Reviewer 4 Report
- English still needs to be improved. Just a few examples from the very first sentences: "Clouds act as a major reflector that the amount of sunlight reflected to space" (a verb is missing in the relative clause). "Clouds play an important role as a reflector that cause[s] change [in] the amount of reflected sunlight in the ultraviolet-visible (UV-Vis) region. Moreover, clouds significantly attenuate[remove the s!] the polarization of the atmosphere". These are just examples, the whole text should be checked by someone who speaks good English;
- All relevant references and previous works should be provided in detail. Reference 23 should be better described;
- Am I to understand that the only innovation compared to the existing algorithms is that only 3 wavelengths are considered in proximity of the oxygen dimer band in place of a slightly larger interval (e.g., 30 nm)? With the computational power of existing PCs, does this really represent a substantial improvement in speed, as mentioned at line 149?
- Replace "albedo" with "surface albedo" in Fig. 2, Table 2 and in the text ("e.g., Sect. 3.2).
